axial conductance; drought; Hagen–Poiseuille; root topology; xylem.

**Corresponding author:**
Alexandre Grondin;
Email: alexandre.grondin@ird.fr

LB and JCBC contributed equally.

**Associate Editor:**
Dr. Poonam Mehra

# Role of xylem in root hydraulics: Functionality and implications for drought adaptation

Luke Barry[1], Juan Carlos Baca Cabrera[2], Mikael Lucas[1], Guillaume Lobet[3], Yann Boursiac[4] and Alexandre Grondin[1]

[1]DIADE, Université de Montpellier, IRD, CIRAD, France; [2]Forschungszentrum Jülich GmbH, Germany; [3]Earth and Life Institute, UCLouvain, Belgium; [4]IPSiM, Université de Montpellier, CNRS, INRAE, Institut Agro, France

## Abstract

Root water transport has been viewed as primarily limited by the radial component, with the axial pathway considered highly conductive and non-limiting. This is supported by theoretical estimates of axial conductance using the Hagen–Poiseuille equation. However, increasing evidence indicates that actual axial conductance is often nearly an order of magnitude lower than predicted, challenging assumptions that it does not limit water uptake. In this review, we discuss how recent model inversion approaches, guided by root hydraulic conductance measurements, have revealed that water transport can be co-limited by radial and axial conductance. We explore possible explanations for this co-limitation, with particular attention to root topology. Finally, we highlight how drought-induced adjustments in xylem vessel traits can reduce axial conductance, contributing to water conservation and cavitation resistance, thereby supporting drought adaptation strategies. Leveraging this overlooked limitation opens new avenues for breeding crops with improved water-use efficiency and resilience to drought .

## 1. Introduction

The vast majority of plant species on Earth possess a vascular system forming a continuous pathway through the roots, stems and leaves that support growth, metabolism and reproduction (Harrison & Morris, 2018). The root vascular system is produced by the procambium originating from the vascular cell initials within the apical root meristems, or by vascular cambium during secondary growth (Scheres et al., 1994). It is composed of xylem vessels responsible for ascending transport of xylem sap, and phloem, which conduct descending phloem sap. In this review, we will focus on xylem vessels that undergo vacuole disruption, cytoplasm clearing, cell wall thickening and lignification to become hollow tube elements, connected to each other or to xylem parenchyma cells through pits (Venturas et al., 2017). At maturity, root xylem vascular tissues represent a network of interconnected pipe-like vessels embedded within a matrix of living and non-living support tissue, responsible for the long-distance transport of water and nutrients, structural support, carbon storage and pathogen resistance (Brodersen et al., 2019).

Ascendant transport of water within the xylem vessels is explained by the cohesion-tension theory in which water flows by capillary forces created by gradients of water potentials across the soil-plant-atmosphere continuum; the higher this gradient, the higher the tension pulling water from the soil within the roots to the shoots (Brown, 2013; Venturas et al., 2017). In roots, water flows radially from the epidermis to the endodermis through two main routes: the apoplastic pathway, which passes through the cell walls, and the cell-to-cell pathways that cross membranes via aquaporins or move symplast to symplast through plasmodesmata (symplastic pathway). After reaching the endodermis, the water enters the stele ground tissue and eventually the xylem vessels, where its flow becomes axial. Recent evidence shows that the root xylem network is complex and vessel number, diameter and connectivity vary between root types and root ages (Clément et al., 2022; Johnson et al., 2014; Wason et al., 2021), as well as sap composition (Zwieniecki et al., 2001). Furthermore, the xylem network, which is embedded within conducting tissues (the outer ground tissue or cortex) of roots from different orders that branch in derivation to each other, may be considered as a functional elementary unit within the global root hydraulic circuit. Therefore, measurements of root axial conductance considering the anatomical and morphological complexities of root segments or of the entire root need to

incorporate the hydraulics of downstream tissues and points of resistance along the hydraulic pathway (Brodersen et al., 2019). Such spatial and temporal considerations have often been overlooked in estimating root axial conductance and may explain some of the discrepancies observed between empirical measurements and theoretical observations of axial water flow within xylem vessels (Jacobsen et al., 2024). Recent developments based on non-invasive imaging and/or multi-scale root modelling provided new insights on the actual contribution of xylem vessels to the hydraulic properties of the root system, challenging the prevailing assumption that axial conductance is non-limiting for root water transport across the entire root system (Bouda et al., 2019; Boursiac et al., 2022; Hacke et al., 2022; Strock et al., 2021).

The root xylem network shapes the hydraulic properties of the root system, playing a key role in plant water use and tolerance to drought (Brodribb, 2009). It was suggested that natural selection led to the complexification of the xylem network in early vascular plants to limit embolism spread and plant hydraulic failure during periods of water limitation (Bouda et al., 2022). Furthermore, modulation of xylem morphology was also linked to native maize domestication along altitude and precipitation gradients in Mexico (McLaughlin et al., 2024). In modern breeding, wheat lines selected for narrower xylem vessel diameters showed improved tolerance to severe drought (Richards & Passioura, 1989) because they slowed down the rate of water use. As drought events increasingly threaten crop yields worldwide, a deeper understanding of how xylem vessels respond to drought and determine potential water flows, both in their development and their role in root water transport as part of a suite of integrated traits, can help identify the drought scenarios where adjusting xylem traits may offer a significant advantage (Vadez et al., 2024). Underlying questions concern the trade-offs associated with xylem adjustment on plant growth, as well as the role of embolism in agricultural fields and its impact on crop productivity.

In this review, we discuss how new model inversion approaches, informed by measurements of root hydraulic conductance, have revealed that water transport in plants can be co-limited by both radial and axial conductance, and explore possible explanations for this co-limitation, with particular attention to root system topology – specifically, the relative arrangement of different root orders and types, and branching patterns within the root system (Fitter, 1987). We further illustrate how adjustments in xylem vessels and axial conductance contribute to water savings and maintenance of hydraulic continuity under drought conditions, thereby enhancing drought tolerance.

## 2. New paradigms related to axial conductance and its contribution to root water flow

The prevailing paradigm in root water transport is that it is principally limited by its radial component. In maize, a striking conclusion has been that the resistance to water flow of a patch of membrane is equivalent to 24 km of xylem vessels of the same diameter (Steudle & Peterson, 1998). Steudle and Peterson (1998) estimated that, when the late metaxylem is mature, the axial resistance of a segment is four orders of magnitude lower than that of the radial direction. Although this may be a general observation among plant species, it may falsely convey the idea that the xylem conductance is not a limiting factor on the overall water transport capacity of a root system. The following sections examine new evidence of axial limitations in root water flow and investigate potential mechanisms responsible for these constraints.

### 2.1. Discrepancies between theoretical and experimental measurements of xylem conductance

The Hagen–Poiseuille equation has been used to estimate xylem conductance and sap flow rate for many decades (Doussan, 1998; Frensch & Steudle, 1989; Landsberg & Fowkes, 1978). In its simplest form, the axial hydraulic conductance ($m^4$ $s^{-1}$ $MPa^{-1}$) of a single vessel element can be calculated by:

$$Lax = \pi \, r^4 \div 8 \, \eta$$

where $r$ is the radius of a xylem vessel (m) and $\eta$ is the viscosity of water ($MPa$ $s^{-1}$). This method has the great advantage of being accessible to a wide range of labs equipped with a microscope, which enables visualization of root cross-sections. However, it has been observed on many occasions that the theoretical Hagen–Poiseuille computation of xylem conductance is larger than the conductance measured experimentally (Bouda et al., 2019; Boursiac et al., 2022; Frensch & Steudle, 1989; Landsberg & Fowkes, 1978). A first example comes from measurements of the axial hydraulic properties of small, unbranched excised root segments using root pressure probes, in which loss of resistance before and after cutting a segment is used to infer the axial conductance of the removed portion (Frensch & Steudle, 1989; Meunier et al., 2018). Using this method, Landsberg and Fowkes (1978) suggested that the actual axial conductance of grass roots was two to three times smaller than the theoretical one. Furthermore, Frensch and Steudle (1989) observed up to a five fold discrepancy in the upper part (>14 cm) of single maize roots, where measured resistance exceeded theoretical estimates.

More recently, the 'Cut and Flow' method was developed by Boursiac et al. (2022) to determine simultaneously both radial and axial conductivities of a fully developed root system. This 'model-assisted' phenotyping combines the use of the root hydraulic architecture model 'Hydroroot' (Boursiac et al., 2022) and measurements of the whole root-system conductance with a pressure chamber (Boursiac et al., 2022). The pressure-induced sap flow of one entire root system was measured in intact plants and after successive cuts from the tips. After reconstitution of the exact architecture, radial and axial hydraulic properties of the RSA were obtained from an optimization procedure of the Hydroroot model parameters in order to match the various sap flow measurements. Boursiac et al. (2022) measured axial conductance values four- to sixfold lower than theoretical estimates along the entire *Arabidopsis* roots and showed through simulations that even a small decrease in xylem conductivity affects sap flow throughout the root system, with varying impact depending on the distance from the base.

Magnetic resonance imaging has also been used to investigate the functional status of the xylem *in planta* (Bouda et al., 2019; Buy et al., 2018). In grapevine, this non-destructive technique was combined with modelling and sap flow measurements to study the xylem network within a stem segment (Bouda et al., 2019). Simulations using Hagen–Poiseuille equation overestimated flow rates in larger vessels and underpredicted it in smaller vessels. This observation is likely to be true for roots as well, highlighting the complexity of xylem conductive tissues.

### 2.2. Influence of topology on axial conductance

Evidence showing that axial conductance is often nearly an order of magnitude lower than theoretical estimates raises the question of whether it is truly non-limiting for root water uptake. In fact, it has

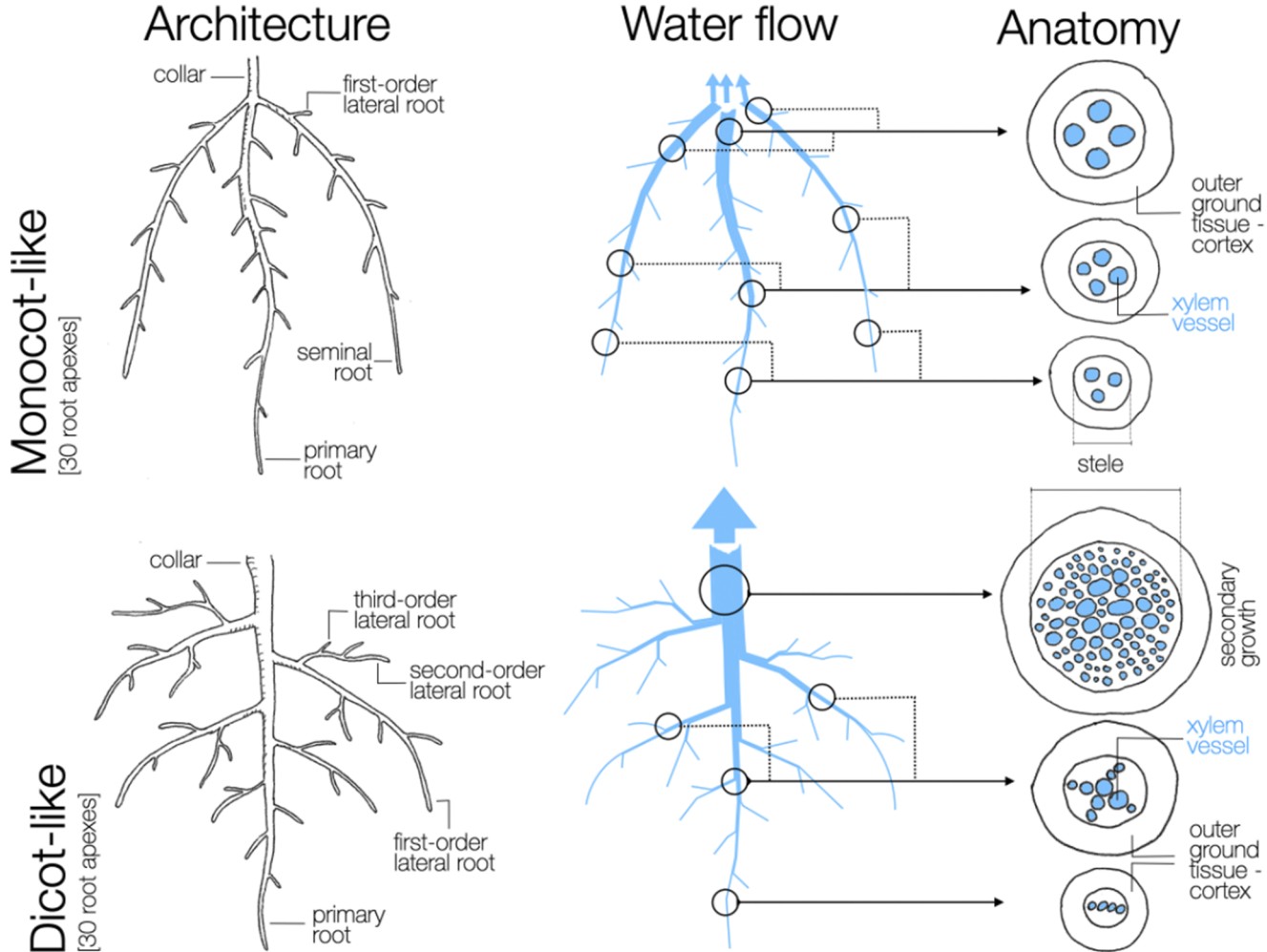

**Figure 1.** Overview of architectural, anatomical and functional differences in terms of water flow between dicot and monocot-like root systems. The anatomical cross-sections represent different maturation stages along the primary root axis.

been proposed that the interplay between xylem vessel morphology and the topological organization of the root system imposes constraints on axial water flow (Bouda et al., 2018). Bouda et al. (2018) showed that, in some cases, root system conductance is more sensitive to axial than radial conductance in absorbing roots, demonstrating that the extent to which axial conductance limits water uptake depends strongly on root network topology and on root length (Figure 1). Furthermore, large differences in xylem water flow have been reported among root types (Meunier et al., 2018), and along the axes of primary roots (Pierret et al., 2006), which can lead to strong variations in water transport limitations depending on the root system architecture. Limitations in axial conductance caused by reductions in metaxylem vessel number and diameter with increasing root depth, for instance, may ultimately influence root water uptake (Clément et al., 2022; Strock et al., 2021).

Comparative analyses of the distinct root architectures of dicot and monocot species offer deeper insights into how axial limitations emerge in different root systems. Typically, dicots (such as soybean in Figure 2) have a taproot system, with a primary root bearing the complete root system. Their secondary roots tend to be long and branched to the second or third order. All water taken up by the root system is funnelled towards the unique primary root,

which must assure its transport to the shoot. Secondary growth and the development of secondary metaxylem in the primary root contribute, in part, to enabling this function. Monocot species (such as wheat in Figure 2) tend to form fibrous root systems. These are composed of a multitude of first-order root axes, originating either from the seed (primary and seminal roots) or shoot nodes (brace and crown roots) that do not undergo secondary growth. They typically bear lateral roots that are relatively short, usually without higher-order roots. In such root systems, water uptake can follow multiple independent pathways to reach the shoot. Therefore, contrast in topology and anatomy between monocot and dicot species are two important factors that may contribute to differences in the relative importance of axial conductance within the whole root system conductance.

To illustrate the effect of these contrasted topologies on the whole root system conductance ($K$rs) and test potential axial conductance limitations to water transport, we simulated root system architectures of a dicot (soybean) and a monocot (wheat) using the whole-plant model CPlantBox (Giraud et al., 2023), with identical segment-scale parametrization of radial ($k_r$) and axial hydraulic ($k_x$) properties (Baca Cabrera et al., 2024; Figure 2 and Supplementary Methods S1). As shown in Meunier et al. (2020) and Baca Cabrera et al. (2024, 2025), $K$rs quickly reaches a maximum value

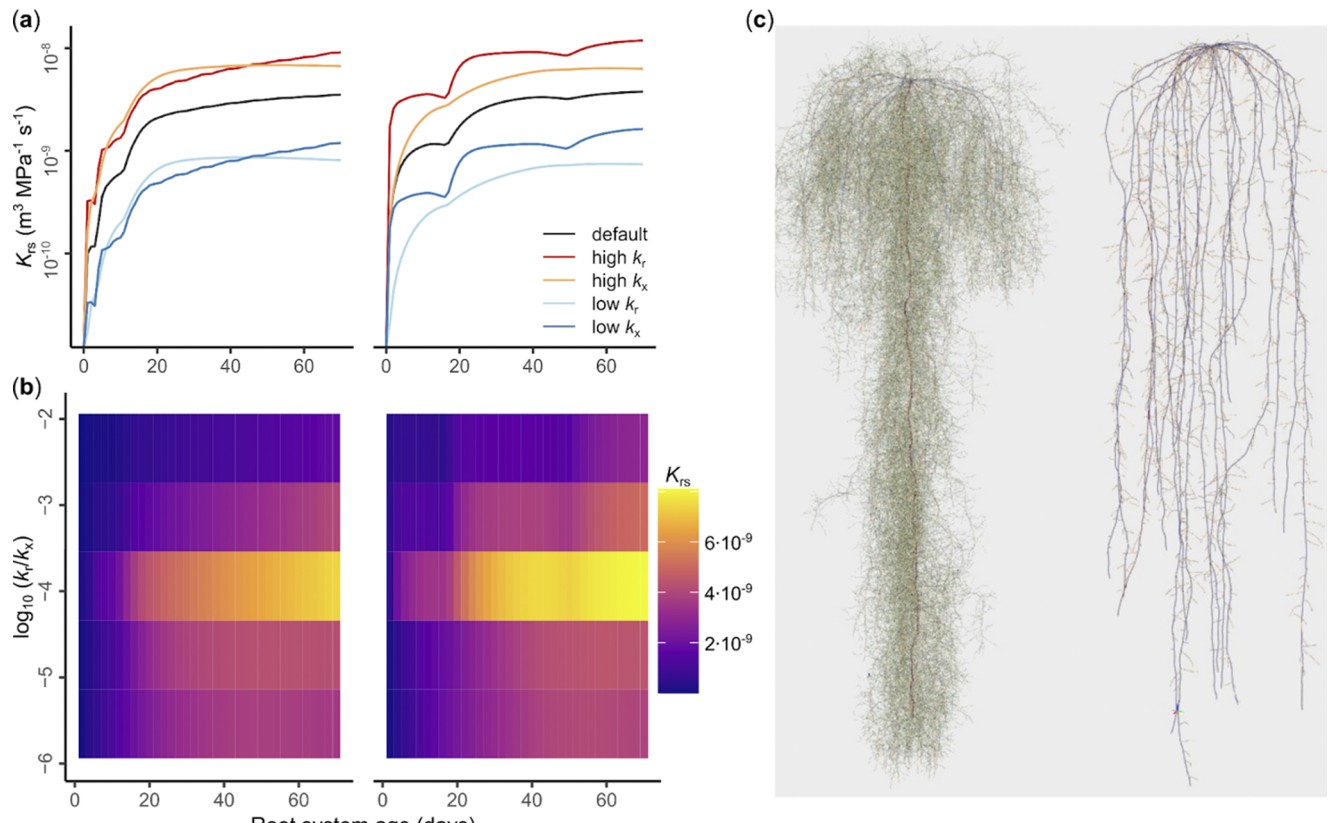

**Figure 2.** The effect of axial conductance ($k_x$) and radial conductivity ($k_r$) changes on whole root system conductance ($K_{rs}$) for dicot (soybean, left panels) and monocot (wheat, right panels) species. (a) Effect of $k_x$ and $k_r$ changes (increase or decrease) on modelled $K_{rs}$ at different root system ages. Based on a default parametrization (Doussan et al., 1998, Baca Cabrera et al., 2024), $k_x$ or $k_r$ were lowered or increased by one order of magnitude for all root types. (b) Heat map with the effect of $k_r/k_x$ ratio changes on $K_{rs}$ at different root system ages (yellow indicates higher, magenta lower $K_{rs}$ values). (c) Contrasting root system architecture at the end of the simulations.

in both species, even though the root system is still growing. A sensitivity analysis was performed by modifying $k_x$ and $k_r$ by an order of magnitude relative to the default parametrization. Interestingly, increasing or decreasing $k_x$ and $k_r$ had a similar impact on $K_{rs}$, challenging the common paradigm that $k_r$ is the predominant limitation to root water uptake (Figure 2a). This effect was particularly pronounced in soybean, where changes in $k_r$ and $k_x$ affected $K_{rs}$ almost identically. In contrast, the effect was less pronounced in wheat, where $k_r$ changes had a stronger influence on $K_{rs}$ than $k_x$ changes, although axial flow limitation was still present. This difference likely reflects that in dicots, where all water must pass through the primary root to reach the shoot, the axial conductance of that root can quickly become a bottleneck. In monocots, changes in $k_x$ have less impact on the root system conductivity, with $k_r$ remaining the main limitation. Since each first-order axis transports only a fraction of water to the shoot, their individual importance is lower. Redundancy root axes and continuous growth maintain overall water uptake capacity.

Additionally, the simulations indicate a non-linear effect of the radial-to-axial conductivity ratio ($k_r/k_x$) on $K_{rs}$. Across both dicot and monocot architectures, the highest $K_{rs}$ values are observed at intermediate $k_r/k_x$ ratios, while lower or higher ratios result in decreased $K_{rs}$ (Figure 2b). These results underscore the importance of a coordinated balance between radial and axial hydraulic properties for optimal water uptake. They are in line with previous studies emphasizing the need for functional integration of both components in determining root water uptake capacity (Bouda et al., 2018), and further challenge the common assumption that

root water uptake is limited primarily by radial conductivity alone.

## 3. The influence of axial conductance on plant drought adaptation

Drought tolerance was associated with a reduced root axial conductance (diameter and/or number of xylem vessels) in several crop species, such as sorghum (Salih et al., 1999), maize (Klein et al., 2020) or wheat (Figure 3a; Hendel et al., 2021; Richards & Passioura, 1989). Reduction in axial conductance within plants experiencing a water deficit during their lifetime has also been observed in wheat (Jafarikouhini & Sinclair, 2023) and rice (Kadam et al., 2017). Interestingly, a switch from metaxylem to protoxylem cell fate upon abscisic acid (ABA) application, along with a subsequent reduction in axial conductance, was observed in dicot species, such as tobacco or tomato (Ramachandran et al., 2021). The molecular mechanisms responsible for this response were elucidated in *Arabidopsis thaliana*, and involve *VASCULAR-RELATED NAC DOMAIN* (*VND*) genes that are induced by ABA (Ramachandran et al., 2021). It has been shown that *VND2* and *VND3* are mainly involved in ABA-mediated enhancement of xylem differentiation rate, while *VND7* mediates a switch in xylem cell fate by altering the secondary cell wall xylem morphology from pitted to spiral or reticulate, the latter being characteristic of protoxylem-like xylem cells (Ramachandran et al., 2021). In parallel, ABA enhances the levels of microRNA165, which acts as a non-cell-autonomous signal to suppress the *HOMEODOMAIN-LEUCINE ZIPPER class III*

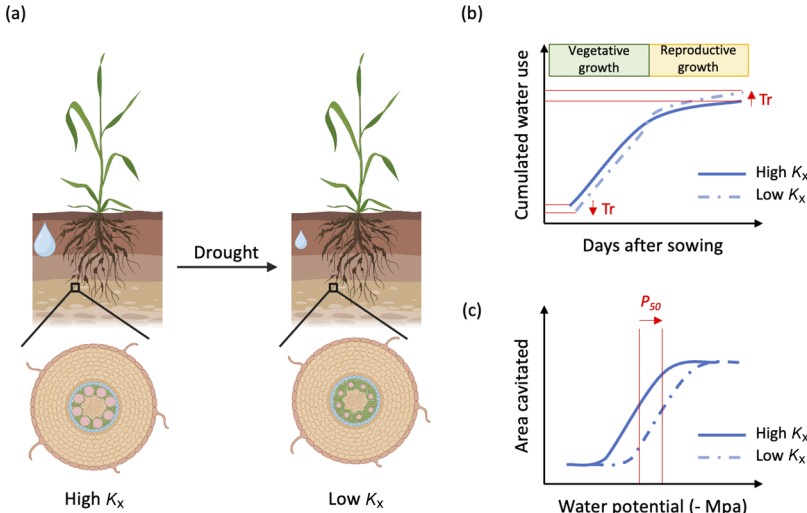

**Figure 3.** Impact of drought stress on axial conductance ($k_x$) and potential implications for plant water use and resistance to cavitation. (a) A common response to drought observed in various species is a reduction in metaxylem diameter, which subsequently decreases axial hydraulic conductance ($k_x$). The figure was created with Biorender.com. (b) This reduction in $k_x$ may support water-saving strategies during the vegetative phase, enabling more conservative water use. As a result, more water may remain available during the reproductive stage, which is critical for reproduction and grain filling. (c) A smaller xylem diameter may also reduce the risk of cavitation. This is because the xylem water potential threshold at which 50% of conductivity is lost due to cavitation tends to become more negative, indicating improved resistance to embolism under water stress.

(*HD-ZIPIII*) transcription factors within the stele, thus promoting protoxylem over metaxylem cell fate (Carlsbecker et al., 2010; Miyashima et al., 2011; for an extensive review, see Cornelis & Hazak, 2022). This switch from metaxylem to protoxylem cell fate presumably helps the plant reduce its water use and vulnerability to cavitation. However, knowledge gaps remain about how such a reduction affects overall hydraulic processes within the soil-plant-atmosphere continuum. The following sections explore the potential physiological significance of reduced root axial conductance and its implications for drought tolerance.

### 3.1. Influence of root axial conductance on plant water use

The plant hydraulic network can be described using a demand and supply scheme, in which the water supply from roots sustains shoot transpiration (Vadez et al., 2024). In this scheme, if the water supply cannot match the water demand, stomatal closure will occur to avoid a large drop in leaf water potential (Tardieu et al., 2017). Stomatal sensitivity to declining leaf water potential has been used to classify plant water-use strategies, distinguishing water savers (or isohydric plants), which close their stomata in response to small drops in leaf water potential, from water spenders (or anisohydric plants), which tolerate larger drops before closure (Tardieu & Simonneau, 1998). It has been hypothesized that reduced xylem diameter and the resulting decline in root axial and overall conductance are associated with water-saving strategies by limiting transpiration more quickly as soil dries or evaporative demand rises, thereby promoting more parsimonious water use and greater drought tolerance (Figure 3b; Burridge et al., 2022; Vadez et al., 2024). In crops such as sorghum, pearl millet or maize, a direct link was established between water savings during the vegetative stage through transpiration restriction in response to the increasing evaporative demand, and yield maintenance under drought (Cooper et al., 2014; Sinclair et al., 2005; Vadez et al., 2013). However, direct evidence linking root hydraulics, and in particular root axial conductance, to whole plant water use remains sparse. This may be explained by the complexity of untangling the effect of root axial conductance from the whole plant hydraulics

through both empirical and modelling approaches as water capture, flow and use integrate multiple architectural, anatomical and functional components operating throughout the plant (Burridge et al., 2022; Klein et al., 2020; Koehler et al., 2023; Strock et al., 2021). Another layer of complexity arises from the interaction between plant hydraulics and the hydraulics of the soil and rhizosphere, with the latter significantly influencing transpiration as the soil dries (Cai et al., 2022; Javaux & Carminati, 2021; Koehler et al., 2022; Sperry et al., 2003). It follows that developing approaches to better understand the crop hydraulic architecture in its environment is important for defining appropriate strategies orienting drought breeding programs based on root hydraulics improvements.

### 3.2. Influence of xylem vessel diameter on cavitation

Increasing capillary tension within the xylem may lead to cavitation damage. Cavitation represents the breaking of intermolecular water bonds that create embolisms within the xylem vessels and block the flow of water (Venturas et al., 2017). Embolism may further spread within the xylem conduit, causing hydraulic network failure and ultimately plant mortality (Mantova et al., 2023). Xylem vulnerability to cavitation is typically measured as the cumulative proportion of xylem conductivity lost versus water tension, resulting in values such as $P_{50}$ that represent the xylem water potential threshold at which 50% of the xylem conductivity is lost due to cavitation (Choat et al., 2012). For instance, the $P_{50}$ in wheat leaves is around −2.87 MPa (Johnson et al., 2018), but it varies between organs, with leaves being more vulnerable than roots and peduncles (Harrison Day et al., 2023). In wheat roots, xylem vulnerability varies largely among root types, with small lateral roots appearing more susceptible to cavitation than larger crown roots (Harrison Day et al., 2023). In wet soils, it was proposed that grasses can generate sufficient positive root pressure to repair xylem embolism overnight, should they experience cavitation during the day due to excessive transpiration (Gleason et al., 2017; Sperry et al., 2003). In drying soils, stomatal closure often precedes substantial losses in xylem conductivity, thereby preventing damage from cavitation (Brodribb & McAdam, 2017; Cochard et al., 2002; Martin-StPaul

et al., 2017). Furthermore, annual crops are usually grown during rainy seasons, soil water potential rarely reaches the permanent wilting point caused by soil hydraulic conductivity loss (Carminati & Javaux, 2020). Therefore, cavitation is generally not considered critical in crops (Corso et al., 2020). Yet, the safety margin, defined as the difference between the minimum midday water potential a plant can experience and the $P_{50}$, is often very narrow (Choat et al., 2012; Franklin et al., 2023). With the increasing frequency of drought and heat stress events, crops are likely to operate closer to this margin more frequently, potentially increasing the risk of cavitation (Brodribb et al., 2020; Buckley, 2005). Decreasing xylem diameter is generally thought to reduce the risk of cavitation (Figure 3c; Jacobsen et al., 2019), although it would also decrease potential water flow sustaining transpiration, hence photosynthesis and growth. Other xylem features, such as xylem length, pit size and density or pit membrane permeability, may also play important roles in resistance to cavitation (Bouda et al., 2019; Brodersen et al., 2013; Venturas et al., 2017). Xylem network organization was linked to a significant increase in stem resistance to embolism spread in grapevines, for instance (Wason et al., 2021). A better understanding of the effects of water deficit on cavitation in crops and the traits potentially affecting cavitation resistance is necessary in future research to clearly define in which drought scenarios hydraulic failure may be problematic for crop productivity. Recent development of non-invasive optical methods for observation of cavitation may represent useful tools for exploring the diversity of xylem vulnerability, potentially contributing to drought adaptation (Brodribb et al., 2016).

## 4. Conclusion and outlook

In root water flow, axial transport has usually been considered less limiting than the radial component, as xylem is often viewed as a simple tube conducting water upward to the shoots. With this pipe-like model, the Hagen–Poiseuille equation is commonly used to estimate axial conductance. However, empirical measurements clearly show that this equation overestimates axial water flow by up to an order of magnitude. This discrepancy may arise from the intrinsic properties of the xylem vessels network, such as its length, branching and connectivity, as well as the overall topology of the root system. Adopting a root hydraulic architectural perspective, both experimental and modelling evidence suggest that, at the scale of the whole root system, radial flow affected by the outer ground tissue and aquaporin activity, and axial flow shaped by the xylem vessels network, may contribute equally to root hydraulic conductance. This aligns with previous studies indicating interactions between root growth and xylem vessel morphology in determining water uptake (Clément et al., 2022; Hendel et al., 2021; Strock et al., 2021), and supports the broader concept that multiple integrated traits enhance plant drought tolerance (Klein et al., 2020).

A typical xylem vessel response to drought involves a reduction in overall root axial conductance, aligning with both natural and human selection for traits that enhance drought adaptation (Bouda et al., 2022; McLaughlin et al., 2024). This conservative response helps reduce water use and increases safety against cavitation. However, it may also compromise transpiration and carbon assimilation (Venturas et al., 2017). Breeding for lower axial conductance could, therefore, introduce trade-offs, particularly by limiting growth and yield under non- or less-stressed conditions. Given the increasing unpredictability of climate and fluctuations in soil moisture throughout the growing season, the question of xylem plasticity, specifically whether it can occur in meristematic zones as well as in mature zones, and whether such plastic responses to drought confer

adaptive benefits without compromising competitiveness under non-stress or transient stress conditions, has become increasingly relevant (Cornelis & Hazak, 2022). In this context, whether xylem plasticity, such as a reduction in vessel size, represents a genuinely adaptive response in all climatic scenarios remains unresolved. Further research is needed to better understand the fitness landscape of xylem plasticity and its value across diverse drought stress scenarios (Schneider & Lynch, 2020).

Developing phenotyping tools to study xylem features and the resulting axial water flow remains a major challenge for exploring trait diversity in crop species and identifying the quantitative loci that control them. Recent advances, such as laser ablation tomography, have significantly increased the throughput of phenotyping root anatomical traits (Strock et al., 2022). When combined with cell-scale hydraulic simulations, these anatomical measurements have enabled the creation of a high-resolution hydraulic conductivity atlas in maize (Heymans et al., 2021). However, integrating architectural, anatomical and cell hydraulic data into genetic analyses, under growth conditions relevant to breeding, remains conceptually and practically difficult. In this context, identifying high-throughput shoot or root traits associated with axial conductance could provide a promising path forward and merit further investigation. Ultimately, overcoming these challenges will be key to better understand xylem limitations in axial water flow, unlocking xylem-related traits for breeding crops better adapted to variable water availability.

**Open peer review.** To view the open peer review materials for this article, please visit http://doi.org/10.1017/qpb.2025.10026.

## Acknowledgements

The authors would like to thank Laurent Laplaze and Vincent Vadez for their critical reading of the manuscript.

**Competing interest.** The authors declare none.

**Data availability statement.** The source code, Python scripts, XML parameter files and simulation outputs supporting this study are publicly available at Zenodo: https://doi.org/10.5281/zenodo.15752385.

**Author contributions.** All authors contributed to the first version of the text and revised the manuscript. JCBC, LB, GL and AG created the figures.

**Funding statement.** The authors acknowledge the financial support from the French Ministry for Research and Higher Education (PhD grant to LB). This work was funded by the Agence Nationale de la Recherche (Plastimil grant ANR-20-CE20-0016 to AG). JCBC is funded by the Deutsche Forschungsgemeinschaft (DFG, German Research Foundation) (SFB 1502/1-2022, Projektnummer: 450058266). GL is supported by the European Union (ERC grant 101125638). YB is supported by Agence Nationale de la Recherche (ANR-22-CE45-0009 EAUDISSECT).

**Supplementary material.** The supplementary material for this article can be found at http://doi.org/10.1017/qpb.2025.10026.

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
