## [Reviewer Report]

I agree with the authors that this is a timtely and important topic.

Just a few minor comments:

In the second paragraph of the introduction, shy isnt symplastic included on the list of water flow pathways along with apoplastic and cell-to-call? Immediately after that statemetn, you state that water moves from the endoermis into the xylem vessels, but doesnt it first move into teh stele tissue?

In the second paragraph of the introduciton you mention ‘the xylem network that is embedded within conducting tissues (the outer cortex) of roots. I find this a bit confusing, the cortex is indeed on the outside of the stele, but the ’outer cortex' sounds like the outer part of the cortex, not just the outer part of the root. I would suggest a minor reword here.

Page 7, ‘usually without lower-order roots’ Should this be ‘higher-order roots’?

---

## [Reviewer Report]

The manuscript by Barry et al. provides a nice review of recent findings on axial conductance in roots, challenging the classical view that radial conductance is the main limitation. The authors highlight that axial conductance can limit water uptake. This is contrary to the common assumption that radial conductance is always the bottleneck. The review effectively integrates anatomical, physiological, and modeling perspectives and emphasizes that the impact of axial conductance depends on root system architecture, including topology, root type and length. The authors also note that both axial (kx) and radial (kr) conductances must be balanced to optimize whole-root water transport. The authors may consider few suggestions to improve the manuscript further:

1. At the beginning, please explain what is meant by root topology and specify the different root features considered, such as branching patterns, root order, and type. This will aid readers who are less familiar with the topic.

2. In Section 2.2, please consider including a table summarizing different factors that lead to differences in axial conductance between dicots and monocots to improve clarity.

3. On page 8, the statement regarding the switch from metaxylem to protoxylem cell fate under ABA in dicots could be expanded to discuss whether this process might protect against embolism.

4. While the review emphasizes biophysical aspects, it would benefit from elaborating on genetic controls of xylem diameter, either in text or as a summary table.

5. Fig. 1. Please expand the legend to clarify what the different cross-sections represent along the root length.

6. Fig. 3. The differences in xylem vessel diameters are not very apparent, and the root/shoot illustrations do not clearly show water stress.The plant exposed to well-watered and drought stress appear visually similar.

7. On page 9, please correct ‘anyisohydric’ to ‘anisohydric’.

---

## [Editor Report]

Dear Authors,

Many thanks for your submission to Quantitative Plant Biology. Both reviewers have appreciated the quality and scope of your review.

We would like to invite you to address a few minor suggestions raised by the reviewers to further improve the manuscript. Please submit a revised version along with a point-by-point response to the reviewers’ comments.

We look forward to receiving your revised manuscript.

Best regards,

Dr. Poonam Mehra